# Data Distillation Can Be Like Vodka: Distilling More Times For Better Quality

**Xuxi Chen*[1], Yu Yang*[2], Zhangyang Wang[1], Baharan Mirzasoleiman[2]**
[1]University of Texas at Austin [2]University of California, Los Angeles
{xxchen,atlaswang}@utexas.edu, {yuyang,baharan}@cs.ucla.edu

## Abstract

Dataset distillation aims to minimize the time and memory needed for training deep networks on large datasets, by creating a small set of synthetic images that has a similar generalization performance to that of the full dataset. However, current dataset distillation techniques fall short, showing a notable performance gap compared to training on the original data. In this work, we are the first to argue that the use of only one synthetic subset for distillation may not yield optimal generalization performance. This is because the training dynamics of deep networks drastically changes during training. Therefore, multiple synthetic subsets are required to capture the dynamics of training in different stages. To address this issue, we propose Progressive Dataset Distillation (PDD). PDD synthesizes multiple small sets of synthetic images, each conditioned on the previous sets, and trains the model on the cumulative union of these subsets without requiring additional training time. Our extensive experiments show that PDD can effectively improve the performance of existing dataset distillation methods by up to $4.3\%$. In addition, our method for the first time enables generating considerably larger synthetic datasets. Our codes are available at https://github.com/VITA-Group/ProgressiveDD.

## 1 Introduction

Dataset distillation aims to generate a very small number of synthetic examples from a large dataset, which can provide a similar generalization performance to that of training on the full dataset (Wang et al., 2018; Loo et al., 2022; Nguyen et al., 2021a;b; Zhou et al., 2022). If this can be achieved, it can significantly reduce the costs and memory requirements of training a deep network on large datasets. Therefore, dataset distillation has gained a lot of recent interest and has found various applications, ranging from continuous learning, neural architecture search, to privacy-preserving ML (Zhao et al., 2021; Dong et al., 2022).

Existing dataset distillation methods generate a set of synthetic examples that match the gradient (Zhao et al., 2021; Zhao & Bilen, 2021b), Neural Tangent Kernel (NTK) (Loo et al., 2022; Nguyen et al., 2021a;b; Zhou et al., 2022), or weights (Kim et al., 2022) of a number of randomly initialized models being trained on the original (Zhao et al., 2021) or synthetic data (Zhao & Bilen, 2021b). However, as matching the entire training dynamics is intractable, existing methods only match the dynamics of *early training iterations*, as short as the first four epochs (Zhao et al., 2021). As the training dynamics of deep networks drastically changes during training, a synthetic subset generated based on the early training dynamics cannot represent the dynamics

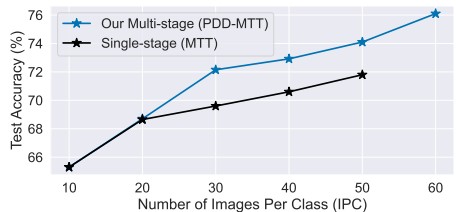

Figure 1: Our multi-stage dataset distillation framework, PDD, improves the state-of-the-art algorithms by iteratively distilling smaller synthetic subsets that together capture longer training dynamics on full data. In the setting shown in the figure, PDD uses MTT as its base distillation method to incrementally generate 5 synthetic subsets of size 10, where each subset captures 15 epochs of training on full data. This yields much better performance compared to generating the same number of images (IPC) in a single stage, i.e., based on the first 15 epochs of training on full data.

---

*The first two authors contributed equally.

of the later training phases. Hence, existing dataset distillation methods suffer from a substantial performance gap to that of training on the original data (Zhao et al., 2021; Kim et al., 2022).

Recent results on optimization and generalization of neural networks revealed that gradient methods have an inductive bias towards learning simple functions, especially early in training (Kalimeris et al., 2019; Hu et al., 2020; Hermann & Lampinen, 2020; Neyshabur et al., 2014; Shah et al., 2020). That is, models trained with (stochastic) gradient methods learn nearly linear functions in the initial training iterations (Kalimeris et al., 2019; Hu et al., 2020). As iterations progress, SGD learns functions of increasing complexity (Kalimeris et al., 2019). This implies that synthetic examples generated based on early training dynamics can only train low-complexity neural networks that perform well on easy examples that are separable by low-complexity models. This limitation is further supported by recent studies (Pooladzandi et al., 2022; Yang et al., 2023), which observed that deep models benefit the most from learning examples of increasing difficulty levels at various training stages and one subset of the training data is not enough to support the entire training.

Building on this observation, to bridge the gap to training on the full data, it is crucial to synthesize examples that can capture the dynamics of later training phases. However, this is very challenging. First, synthesizing examples that match the training dynamics of many randomly initialized networks over longer training intervals has a very high computational cost. Moreover, capturing complex training dynamics over longer intervals requires synthesizing more images, which makes it prohibitively expensive. Finally, even if a larger subset can be generated to match the dynamics of a longer training interval, it is not enough to bridge the gap to training on the full data.

In this work, we address the above challenges by proposing a Progressive Dataset Distillation (`PDD`) pipeline. We are the first to employ a multi-stage idea: specifically, to generate multiple synthetic subsets that can capture the training dynamics in different phases. To do so, we synthesize examples that can capture the training dynamics of the full data in a given training interval. Then, we train the model on the synthesized examples and generate another set of synthetic examples that, together with the previous synthetic sets, capture training dynamics of the full data in the consecutive training interval. Importantly, our progressive distillation approach captures the training dynamics of neural networks more effectively, resulting in superior generalization performance. Figure 1 confirms that by distilling the dynamics of the later training stages on CIFAR-10, `PDD` effectively improves the performance when training on the distilled data.

Our extensive experiments confirm that our multi-stage distillation approach outperforms existing methods by up to $5\%$ on ConvNet and $5\%$ for cross-architecture generalization to ResNet-10 and ResNet-18. Remarkably, `PDD` is the first method to enable generating larger synthetic datasets. In doing so, it considerably bridges the gap to training on the full data by achieving 90% of the full accuracy with only $5\%$ of the full data size on CIFAR-10 and $8\%$ of full data size on CIFAR-100 (Krizhevsky et al., 2009) and provides state-of-the-art accuracy on Tiny-ImageNet (Le & Yang, 2015). We also conduct studies showing that: 1) our multi-stage synthesis framework achieves consistent improvement if new intervals are introduced, which confirms its ability to serve as an effective base method for further condensation (Liu et al., 2023a) where the budgets (*i.e.,* number of images) are subjected to change; 2) our framework generates synthetic samples with strong generalization ability across various architectures; 3) the distillation process can be performed on progressively challenging subsets of the full data at each stage, resulting in minimal performance degradation.

## 2 RELATED WORKS

Dataset Distillation (`DD`) (Wang et al., 2018) aims to generate a synthetic subset from a large training data that can achieve a similar generalization performance to that of training on the full dataset, when trained on. To achieve this, `DD` adopted an optimization process comprising two nested loops. The inner loop involves training a model using the synthesized data until it reaches convergence, while the outer loop aims to optimize the synthetic data such that the trained model generalizes well on the original dataset. More recent studies (Loo et al., 2022; Nguyen et al., 2021a;b; Zhou et al., 2022) leverage the same framework but use kernel methods, such as Neural Tangent Kernel (NTK), to approximate the inner optimization in a closed form. While kernel-based algorithms achieved higher accuracy than `DD` (Wang et al., 2018) on networks that satisfy the infinite-width assumption, they do not work well in practice as the constant NTK assumption does not generally hold.

Another set of methods relies on gradient matching. In particular, `DC` (Zhao et al., 2021) minimizes the distance between the gradients of the synthetic and original data on the network being trained

on the synthetic data. `DSA` (Zhao & Bilen, 2021b) improves upon `DC` by applying differentiable siamese augmentations to both the original and synthetic data while matching their training gradients. Incorporating differentiable data augmentation has been adopted by almost all subsequent studies. Later on, `IDC` (Kim et al., 2022) proposed a multi-formulation framework to generate more augmented examples from the same set of synthetic data, to boost the performance with the same storage budget. The synthetic data is generated by minimizing the distance between the gradients of the synthetic and original data on the network being trained on the full data. Concurrent to our work, Feng et al. (2024) proposes to unroll gradients within a randomly picked small window along the training trajectory and generate nested synthetic datasets of different IPCs.

Besides matching the gradients, other methods involve matching the training trajectories of the network parameters (Cazenavette et al., 2022) or the data distribution (Wang et al., 2022; Zhao & Bilen, 2023). `MTT` (Cazenavette et al., 2022) pre-computes and stores training trajectories of expert networks trained on the original data, and then minimizes the distance between the parameters of the network trained on the synthetic data and the expert networks. `CAFE` (Wang et al., 2022) matches the features between the synthetic and real data in all intermediate layers. To avoid the expensive bi-level optimization, `DM` (Zhao & Bilen, 2021a) minimizes the distance between feature embeddings of the synthetic and real data based on randomly initialized networks. More recently, `HuBa` (Liu et al., 2022) proposed to distill a dataset into two components, Bases and Hallucination to increase the representation capability of distilled datasets. `IT-GAN` (Zhao & Bilen, 2022) inverted the training samples into latent spaces and further fine-tuned towards a distillation objective, `GLaD` (Cazenavette et al., 2023) used generative adversarial networks as a prior to help the cross-architecture generalization. More recently, `DREAM` (Liu et al., 2023b) selects representative samples during matching to improve the performance and efficiency of dataset distillation, and `DATM` (Guo et al., 2024) aligns the difficulty of the generated patterns with the size of the synthetic dataset.

Most existing works generate a set of synthetic examples that match the dynamics of neural networks during early-training stage or at multiple random initializations. In contrast, we show that progressively generating multiple synthetic subsets to match the training dynamics in different stages of training yields superior performance.

## 3 PROBLEM FORMULATION AND PRELIMINARY

Given a large dataset $\mathcal{T} = \{(\boldsymbol{x}_i, y_i)\}$ consisting of $|\mathcal{T}|$ samples from $C$ classes, data distillation aims to learn a synthetic set $\mathcal{S} = \{(\boldsymbol{s}_i, y_i)\}$ with synthetic samples $|\mathcal{S}|$ so that deep neural networks can be trained in $\mathcal{S}$ and achieve comparable generalization performance to those trained on $\mathcal{T}$. Formally,

$$\mathbb{E}_{\boldsymbol{x} \sim P(\mathcal{D})}[\mathcal{L}(\phi_{\boldsymbol{\theta}^{\mathcal{T}}}(\boldsymbol{x}), y)] \simeq \mathbb{E}_{x \sim P(\mathcal{D})}[\mathcal{L}(\phi_{\boldsymbol{\theta}^{\mathcal{S}}}(\boldsymbol{x}), y)], \tag{1}$$

where $P(\mathcal{D})$ is the real data distribution, $\phi_{\boldsymbol{\theta}^{\mathcal{T}}}(.)$ and $\phi_{\boldsymbol{\theta}^{\mathcal{S}}}(.)$ are models trained on $\mathcal{T}$ and $\mathcal{S}$ respectively. $\mathcal{L}(.,.)$ is the loss function, e.g., cross-entropy loss.

State-of-the-art dataset distillation methods condense the real dataset into a small synthetic set by matching the gradient of full data along the synthetic or real trajectory, expressed as follows:

$$\arg\min_{\mathcal{S}} \mathbb{E}_{\boldsymbol{\theta}_0 \sim P_{\boldsymbol{\theta}_0}}[\sum_{t=0}^{T-1} D(\nabla_{\boldsymbol{\theta}}\mathcal{L}^{\mathcal{S}}(\boldsymbol{\theta}_t), \nabla_{\boldsymbol{\theta}}\mathcal{L}^{\mathcal{T}}(\boldsymbol{\theta}_t))], \tag{2}$$

where $\boldsymbol{\theta}_t$ denotes the model parameters, and $D$ computes distance between the gradients. `DC` (Zhao et al., 2021) and `DSA` (Zhao & Bilen, 2021b) update $\boldsymbol{\theta}_t$ by minimizing the loss $\mathcal{L}^{\mathcal{S}}(\boldsymbol{\theta}_t)$ on the synthetic data. `IDC` (Kim et al., 2022) showed that updating $\boldsymbol{\theta}_t$ by minimizing the loss $\mathcal{L}^{\mathcal{T}}(\boldsymbol{\theta}_t)$ on the full data yields superior performance. Matching the gradient of the augmented version of the training and synthetic examples further improves the performance (Zhao et al., 2021; Kim et al., 2022).

Alternatively, `MTT` (Cazenavette et al., 2022) trains two models on synthetic and real data and matches weight trajectories $\boldsymbol{\theta}_{t+N}$ of length $N$ when training the model on synthetic data $\mathcal{S}$ with weight trajectories $\boldsymbol{\theta}_{t+M}^*$ of length $M \gg N$ when training the model on real data $\mathcal{T}$:

$$\arg\min_{\mathcal{S}} \frac{\|\boldsymbol{\theta}_{t+N} - \boldsymbol{\theta}_{t+M}^*\|_2^2}{\|\boldsymbol{\theta}_t - \boldsymbol{\theta}_{t+M}^*\|_2^2}. \tag{3}$$

Existing dataset distillation methods synthesize examples based on the gradients or weights of the models during the initial training epochs (Cazenavette et al., 2022; Kim et al., 2022), or match outputs

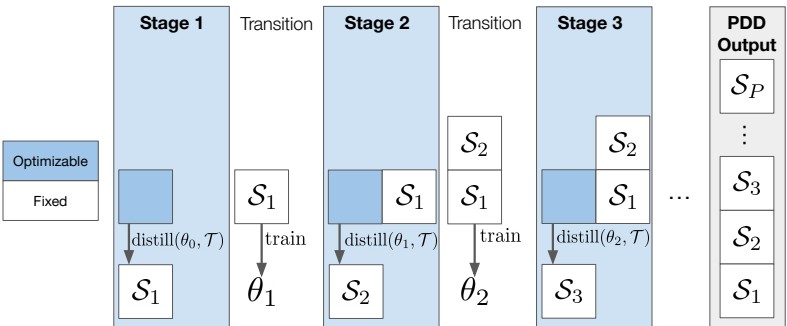

Figure 2: An illustration of the proposed Progressive Dataset Distillation (PDD) framework. It consists of multiple distillation stages and transitions in between. In each distillation stage, we distill a new set of images conditioned on images synthesized in the previous stages. In transitions, we train models on all synthesized images so far, as the starting weights for the next distillation stage to capture longer training dynamics. Our framework can be applied to any dataset distillation algorithm.

of multiple randomly initialized models (Zhao & Bilen, 2021a). The most successful methods, synthesize examples that capture the training dynamics of models trained on *full data* (Kim et al., 2022; Zhao et al., 2021). However, they only capture the *early* training dynamics. For example, IDC (Zhao et al., 2021) and MTT (Kim et al., 2022) synthesize examples by matching gradients and weights of the first 4 and 15 epochs of a 200 training pipeline respectively, when distilling CIFAR-10 and CIFAR-100. This is because matching weights or gradients over longer intervals becomes computationally difficult and does not yield high-quality synthetic data. This introduces a performance gap to that of training on the original data.

## 4 PROGRESSIVE DATASET DISTILLATION (PDD)

Next, we introduce our Progressive Dataset Distillation (PDD) framework to generate synthetic images that match the training dynamics of different stages of training.

### 4.1 DISTILLING MULTIPLE TRAINING STAGES

To capture the learning dynamics of different training stages, our key idea, shown in Figure 2, is to generate a *sequence* of small synthetic datasets $\mathcal{S}_1, \mathcal{S}_2, \cdots, \mathcal{S}_P$, so that each synthetic dataset $\mathcal{S}_i$ captures the training dynamics in the full data in a different stage of training. Then at test time, when the model is trained on the synthetic images, we can train the model on different subsets to mimic different stages of training on full data.

However, naively dividing the full training pipeline into $P$ intervals and generating a subset based on the training dynamics of each interval does not yield a satisfactory performance, due to the following reasons. First, generating different synthetic subsets independently results in capturing redundant information in the subsets and does not improve the performance. Second, since the synthetic sub-

---

**Algorithm 1** Progressive Dataset Distillation (PDD)

**Input:** A dataset distillation algorithm $\mathcal{A}$, full training set $\mathcal{T}$
**Output:** Model trained on a series of synthetic datasets: $\mathcal{S}_1, \mathcal{S}_2, \ldots, \mathcal{S}_N$
**Generating synthetic subsets: PDD**
$\mathcal{S}_0 \leftarrow \emptyset$
Initialize $\boldsymbol{\theta}_0$ randomly
**for** $i = 1, 2, \ldots, P$ **do**
    $\mathcal{S}_i = \mathcal{A}(\boldsymbol{\theta}_i, \mathcal{T} | \cup_{j=1}^{i-1} \mathcal{S}_j)$
    $\boldsymbol{\theta}_i = \arg\min_{\boldsymbol{\theta}} \mathcal{L}(\boldsymbol{\theta}, \cup_{j=1}^{i-1}\mathcal{S}_j, \boldsymbol{\theta}_{i-1})$
**end for**
**Evaluation: Training on the PDD subsets**
Initialize $\boldsymbol{\theta}_0$ randomly
**for** $i = 1, 2, \cdots, P$ **do**
    $\boldsymbol{\theta}_i = \arg\min_{\boldsymbol{\theta}} \mathcal{L}(\boldsymbol{\theta}, \cup_{j=1}^{i-1}\mathcal{S}_j, \boldsymbol{\theta}_{i-1})$
**end for**

---

sets are small, at test time when the model is trained on the synthetic images, minimizing the loss on subset $\mathcal{S}_{i+1}$ results in forgetting the previously learned information from subsets $\mathcal{S}_1, \cdots \mathcal{S}_i$. Finally, even if forgetting can be prevented, transitioning from training on $\mathcal{S}_i$ to training on $\mathcal{S}_{i+1}$ at test time changes the training loss and interrupts the optimization pipeline. This does not allow the model to learn well from multiple synthetic subsets.

To address the above issues, we synthesize each subset $\mathcal{S}_i$ based on the dynamics of training on the full data at stage $i$, *conditioned* on the previous subsets $\mathcal{S}_1, \mathcal{S}_2, \cdots, \mathcal{S}_i$. That is, we generate $\mathcal{S}_i$ such that $\mathcal{S}_1 \cup \mathcal{S}_2 \cup \cdots \cup \mathcal{S}_i$ captures the training dynamics at stage $i$. Note that we only synthesize $\mathcal{S}_i$ at interval $T_i$ while keeping $\mathcal{S}_1, \mathcal{S}_2, \cdots, \mathcal{S}_{i-1}$ fixed. This prevents capturing redundant information in subset $\mathcal{S}_i$ that are already captured by previous subsets $\mathcal{S}_1, \mathcal{S}_2, \cdots, \mathcal{S}_{i-1}$. Next, to address the discontinuity in training on multiple subsets, we synthesize every subset $\mathcal{S}_i$ based on the training dynamics of full data starting from parameters $\boldsymbol{\theta}_i$, where training on $\mathcal{S}_1 \cup \mathcal{S}_2 \cup \cdots \cup \mathcal{S}_{i-1}$ is finished. This allows smooth transitioning between different subsets when training on the synthetic data. Finally, at test time when the model is trained on the synthetic subsets, to prevent forgetting the information learned from the previous subsets, we first train the model on $\mathcal{S}_1$, then $\mathcal{S}_1 \cup \mathcal{S}_2$, and keep training on the union of the previous subsets in addition to the new one $\mathcal{S}_1 \cup \mathcal{S}_2 \cup \cdots \cup \mathcal{S}_i$.

We summarize our pipeline in Algorithm 1. Formally, for $i = 1, \cdots, P$, we generate a synthetic subset $\mathcal{S}_i$ as follows:

$$\mathcal{S}_i = \mathcal{A}(\boldsymbol{\theta}_i, \mathcal{T} \,|\, \cup_{j=1}^{i-1} \mathcal{S}_j) \qquad \text{s.t.} \qquad \boldsymbol{\theta}_i = \arg\min_{\boldsymbol{\theta}} \mathcal{L}(\boldsymbol{\theta}, \cup_{j=1}^{i-1} \mathcal{S}_j, \boldsymbol{\theta}_{i-1}), \tag{4}$$

where $\mathcal{L}(\boldsymbol{\theta}, \mathcal{S}, \boldsymbol{\theta}_{i-1})$ is the loss of the model trained on data $\mathcal{S}$ starting from $\boldsymbol{\theta}_{i-1}$. $\mathcal{A}$ can be any dataset distillation method, such as `DC` (Zhao et al., 2021), `DSA` (Zhao & Bilen, 2021b), `IDC` (Zhao et al., 2021), and `MTT` (Kim et al., 2022), described in Equation 2 and 3.

**Distillation and training costs** Note that conditioning the distillation on previous subsets does not increase the cost of synthesizing a new subset, as we generate the same number of synthetic images at every interval. On the other hand, at test time, we train on fewer images in total. This is because instead of training on $k = |\mathcal{S}|$ synthetic examples during the entire training, `PDD` with $m$ intervals first trains the model on $k/m$ synthetic images. Then, it trains the model on $2k/m$ synthetic images and keeps increasing the number of training examples until it trains on $k$ examples at the final interval.

### 4.2 Discarding Easier-to-learn Examples at Later Stages

As training progress, `PDD` generates synthetic examples that enable the network to learn higher complexity functions. This implies that at later stages, we can safely discard the examples that are learned early in training with lower-complexity functions from the distillation pipeline. To calculate the learning difficulty of examples, we use the forgetting score (Toneva et al., 2019) defined as the number of times the prediction of every example changes from correct to wrong during the training. Examples with higher forgetting scores are learned later during the training with higher complexity functions. On the other hand, examples that have a very low forgetting score are those that can be classified by lower complexity functions, early in training. At every distillation stage, we drop examples with low forgetting scores and focus the distillation on examples with increasing levels of difficulty, measure by forgetting score. This improves the efficiency of `PDD` without harming the performance, as we will confirm experimentally.

Next, we will show experimentally that `PDD` effectively trains higher-quality neural networks with superior generalization performance without increasing the training time on the synthetic examples.

## 5 Experiments

In this section, we assess the classification performance of neural networks trained on synthetic images generated by our framework. In addition to evaluating on the architecture used for distillation, we also investigate the transferability of the distilled images to larger models with different architectures. We further show with ablation studies that `PDD` trains models with increasing classification accuracy when we increase the number of intervals, and confirm the importance of conditioning and transitions.

### 5.1 Experimental Settings

**Datasets.** We conduct our experiments on three datasets: CIFAR-10, CIFAR-100 (Krizhevsky et al., 2009) and Tiny-ImageNet (Le & Yang, 2015). CIFAR-10 and CIFAR-100 consist of $50,000$ training images, with 10 and 100 classes, respectively. The image size for CIFAR is $32 \times 32$. Tiny-ImageNet contains $100,000$ training images from 200 categories, with a size of $64 \times 64$.

**Baselines.** We consider both data selection and distillation algorithms as baselines, including random selection, Herding (Welling, 2009), K-center (Farahani & Hekmatfar, 2009), and Forgetting (Toneva et al., 2019) for selection and `DC` (Zhao et al., 2021), `DSA` (Zhao & Bilen, 2021b), `DM` (Zhao & Bilen,

Table 1: Test accuracy of ConvNets on CIFAR-10/100 and Tiny-ImageNet, trained on synthetic samples generated by various models with different numbers of images per class (IPC). Our algorithm (PDD) improves upon baseline methods through its multi-stage distillation pipeline, narrowing the performance gap relative to training on the full dataset. PDD results are reported for 5 stages.

| | Dataset | CIFAR-10 | | CIFAR-100 | | Tiny-ImageNet | |
|---|---|---|---|---|---|---|---|
| | IPC | 10 | 50 | 10 | 50 | 10 | 50 |
| Selection | Random | $26.0 \pm 1.2$ | $43.4 \pm 1.0$ | $14.6 \pm 0.5$ | $30.0 \pm 0.4$ | $5.0 \pm 0.2$ | $15.0 \pm 0.4$ |
| | Herding | $31.6 \pm 0.7$ | $40.4 \pm 0.6$ | $17.3 \pm 0.3$ | $33.7 \pm 0.5$ | $6.3 \pm 0.2$ | $16.7 \pm 0.3$ |
| | K-Center | $14.7 \pm 0.9$ | $27.0 \pm 1.4$ | $7.1 \pm 0.2$ | - | - | - |
| | Forgetting | $23.3 \pm 1.0$ | $23.3 \pm 1.1$ | $15.1 \pm 0.2$ | $30.5 \pm 0.3$ | $5.1 \pm 0.2$ | $15.0 \pm 0.3$ |
| Distillation | DC | $44.9 \pm 0.5$ | $53.9 \pm 0.5$ | $25.2 \pm 0.3$ | - | - | - |
| | DSA | $52.1 \pm 0.5$ | $60.6 \pm 0.5$ | $32.3 \pm 0.3$ | $42.8 \pm 0.4$ | - | - |
| | CAFE | $46.3 \pm 0.6$ | $55.5 \pm 0.6$ | $27.8 \pm 0.3$ | $37.9 \pm 0.3$ | - | - |
| | CAFE + DSA | $50.9 \pm 0.5$ | $62.3 \pm 0.4$ | $31.5 \pm 0.2$ | $42.9 \pm 0.2$ | - | - |
| | DM | $48.9 \pm 0.6$ | $63.0 \pm 0.4$ | $29.7 \pm 0.3$ | $43.6 \pm 0.4$ | $12.9 \pm 0.4$ | $24.1 \pm 0.3$ |
| | MTT | $65.3 \pm 0.7$ | $71.9 \pm 0.2$ | $39.6 \pm 0.3$ | $47.7 \pm 0.2$ | $23.2 \pm 0.3$ | $28.0 \pm 0.3$ |
| | **PDD**+MTT | $\mathbf{66.9 \pm 0.4}$ | $\mathbf{74.2 \pm 0.5}$ | $\mathbf{43.1 \pm 0.7}$ | $\mathbf{52.0 \pm 0.5}$ | $\mathbf{27.3 \pm 0.5}$ | $\mathbf{29.2 \pm 0.6}$ |
| | IDC | $67.5 \pm 0.5$ | $74.5 \pm 0.1$ | $45.1 \pm 0.3$ | $52.5 \pm 0.4$ | - | - |
| | **PDD**+IDC | $\mathbf{67.9 \pm 0.2}$ | $\mathbf{76.5 \pm 0.4}$ | $\mathbf{45.8 \pm 0.5}$ | $\mathbf{54.4 \pm 0.4}$ | - | - |
| Full Data | | 88.1 | | 56.2 | | 37.6 | |

2021a), CAFE (Wang et al., 2022), IDC (Kim et al., 2022), and MTT (Cazenavette et al., 2022) for distillation. Herding (Welling, 2009) greedily selects samples to approximate the mean of the entire dataset; Forgetting score (Toneva et al., 2019) keeps track of how many times a training sample is learned and forgotten during the training and keeps examples with the highest forgetting score; K-Center (Farahani & Hekmatfar, 2009) selects the samples to minimize the maximum distance between a data point and its center. Distillation baselines are introduced in Section 2.

**Architectures.** Our experimental settings follow that of Cazenavette et al. (2022): we employ a ConvNet for distillation, with three convolutional blocks for CIFAR-10 and CIFAR-100 and four convolutional blocks for Tiny-ImageNet, each containing a 128-kernel convolutional layer, an instance normalization layer (Ulyanov et al., 2016), a ReLU activation function (Nair & Hinton, 2010) and an average pooling layer. We include ResNet-18 and ResNet-10 (He et al., 2016) to assess the transferability of the synthetic images to other architectures.

**Distillation Settings.** We adopt two representative baseline methods on which we apply our framework: IDC and MTT, which are widely used state-of-the-art dataset distillation methods. During the matching process, we adopt the optimal hyper-parameter reported in the original paper of each dataset distillation method in each stage of PDD without further tuning. We report the number of images PDD distills at each stage and also report the number of synthetic sets $P$ in our results to enable a comparison between PDD and the baselines. Note that the number of synthetic sets has a monotonic effect on the models' testing accuracies.

**Evaluation.** Once the synthetic subsets have been constructed for each dataset, they are used to train randomly initialized networks from scratch, followed by evaluation on their corresponding testing sets. For PDD, we sequentially train models after each interval on all synthetic samples that have already been generated up to the current interval. For each experiment, we report the mean and the standard deviation of the testing accuracy of 5 trained networks. To train networks from scratch at evaluation time, we use the SGD optimizer with a momentum of 0.9 and a weight decay of $5 \times 10^{-4}$. For IDC, the learning rate is set to be 0.01. For MTT, the learning rate is simultaneously optimized with the synthetic images. During the evaluation time, we follow the augmentation strategies of each method to train networks from scratch.

## 5.2 EVALUATING DISTILLED DATASETS

**Setup.** We demonstrate the effectiveness of the proposed multi-stage distillation by applying PDD to MTT and IDC to distill CIFAR-10/100 and Tiny-ImageNet. Table 1 compares PDD with state-of-the-art baselines for different values of Images Per Class (IPC) distilled in 5 stages. We specify baselines' IPC and PDD's IPC to be 10 and 50 for all the benchmarks. For Tiny-ImageNet, we only conduct experiments with MTT as IDC's distillation time is prohibitively expensive in this higher resolution. Based on the default settings, single-stage IDC distills 4 epochs of training on the real images; MTT distills 15 epochs for CIFAR-10, 20 for CIFAR-100, and 40 for Tiny-ImageNet.

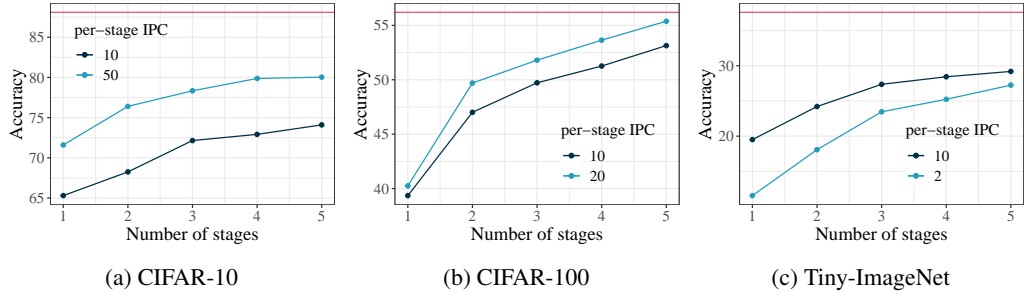

| (a) CIFAR-10 | (b) CIFAR-100 | (c) Tiny-ImageNet |

Figure 3: ConvNets' test accuracy on CIFAR-10, CIFAR-100 and Tiny-ImageNet after training on samples distilled by `PDD` + `MTT` with multiple stages with larger per-stage IPCs. Left: performance on CIFAR-10; Middle: performance on CIFAR-100; Right: performance on Tiny-ImageNet. The red lines indicate the performance of training on respective full data.

**Comparing to Single Stage Distillation.** We see that `PDD` consistently improves the performance across all data selection and distillation baselines with the same IPCs, especially when we distill longer training dynamics (i.e., 15 epochs with `MTT`) on the real images in each stage. Specifically, `PDD` + `MTT` outperforms `MTT` by significant margins of $1.6\%/2.3\%$ on CIFAR-10 IPC-10/50, $3.5\%/4.3\%$ on CIFAR-100, and $4.1\%/1.2\%$ on Tiny-ImageNet. After applying `PDD` to `IDC`, we witness an substantial improvement on performance across different datasets: $0.4\%/2.0\%$ on CIFAR-10 IPC-10/50, respectively, and $0.7\%/1.9\%$ on CIFAR-100 IPC-10/50, respectively.

**Scaling up synthetic datasets: towards bridging the gap to training on the full data.** In Figure 3a and 3b, we extend our experiments with `MTT` by maintaining a constant per-stage IPC while progressively increasing the number of stages. This setting enables scaling of synthesis process to generate larger total IPC, because the images generated in earlier stages are employed in subsequent stages. We conduct these experiments on CIFAR-10 and CIFAR-100, respectively, and set the per-stage IPC to 10/50 for CIFAR-10, 10/20 for CIFAR-100, and 2/10 for Tiny-ImageNet. Remarkably, `PDD` considerably bridges the gap to training on the full dataset by achieving 90% of the full accuracy with only 5% of the full data size on CIFAR-10 (which means that IPC = 250) and 10% of full data size on CIFAR-100 (which means that IPC = 50). Notably, for CIFAR-100, we utilize 20% of the complete dataset, resulting in an IPC value of 100, yet achieve a comparable performance. On Tiny-ImageNet, applying `PDD` with `MTT` could also reach 80% of the performance obtained by training on the full data after distilling 50 images per class.

## 5.3 CROSS-ARCHITECTURE GENERALIZATION

Next, we evaluate the generalization performance of `PDD` on architectures that are different from the one we used to distill CIFAR-10. Following the settings of cross-architecture experiments in the original papers, we use batch normalization layers when evaluating on `IDC`, and use instance normalization layers for `MTT`. We follow the same evaluation pipeline for each baseline method to acquire and present the test accuracy in Table 2.

Images distilled by `PDD` improves other architectures' performance ($1.7\%/1.5\%$ on ResNet-10 and $1.8\%/0.9\%$ on ResNet-18) when using IPC

Table 2: Performance on other architectures of networks with synthetic datasets generated on ConvNets by baselines vs. `PDD` + baselines.

| (Total) IPC | Method | ResNet-10 | ResNet-18 |
|---|---|---|---|
| 10 | IDC | $63.0 \pm 0.6$ | $63.6 \pm 0.4$ |
| | **PDD**+IDC | $\mathbf{63.2 \pm 1.4}$ | $\mathbf{63.9 \pm 0.6}$ |
| | MTT | $46.4 \pm 0.4$ | $45.2 \pm 0.3$ |
| | **PDD**+MTT | $\mathbf{47.1 \pm 0.3}$ | $\mathbf{46.0 \pm 0.4}$ |
| 50 | IDC | $70.7 \pm 0.6$ | $69.8 \pm 0.3$ |
| | **PDD**+IDC | $\mathbf{72.4 \pm 0.4}$ | $\mathbf{71.6 \pm 0.3}$ |
| | MTT | $63.1 \pm 0.4$ | $62.6 \pm 0.4$ |
| | **PDD**+MTT | $\mathbf{64.6 \pm 0.9}$ | $\mathbf{63.5 \pm 0.5}$ |

= 50, and show considerable improvement ($0.2\%/0.7\%$ on ResNet-10 and $0.3\%/0.8\%$ on ResNet-18) compared to using the single-stage `MTT` and `IDC` when the total IPC is 10. These results indicate that our distilled images from multiple stages are robust to changes in network architectures.

## 5.4 ABLATION STUDIES

**Effect of Progressive Training.** When training a model on the $P$ synthetic subsets, `PDD` progressively trains on the union of the first $i$ synthetic sets, for $i = 1, \cdots, P$. To demonstrate the effectiveness of this progressive way of training, we explore multiple choices of training pipelines with the `PDD` generated synthetic sets: (1) *Union*: we train on the union of the synthetic sets generated

in all $P$ stages, i.e., $\cup_{j=1}^{P} \mathcal{S}_j$; (2) *Sequential*: we train on different $\mathcal{S}_i$ in the order they are generated; (3) *Progressive*: we progressively train on union of the first $i$ synthetic sets, i.e., $\cup_{j=1}^{i} \mathcal{S}_j$.

Table 3 compares the above training methods when evaluating the synthetic sets `PDD` distilled for CIFAR-10 with a fixed per-stage IPC = 10 and different numbers of stages $P$. For all the base distillation algorithms, namely `MTT` and `IDC`, progressive training is consistently better than union and outperforms sequential training with a large margin in particular for larger $P$. This confirms the necessity of progressive training to prevent forgetting the previously learned information. Note that `PDD + MTT` performs poorly with the union pipeline because `MTT` learns the learning rate for each set of synthetic images, so a single learning rate is not suitable for training on the union.

**Importance of transitions and conditioning.** There are two key designs in `PDD` that are essential for the success of multi-stage dataset distillation: (1) *transition* between stages by generating a new synthetic subset based on the training trajectory starting from the point where training on the union of the previous synthetic subsets is finished; and (2) *conditioning* on synthetic images distilled in earlier stages when generating a new synthetic set for the current training stage.

Table 3: Effect of training on `PDD` distilled subsets. Testing accuracy on CIFAR-10 after being trained on 10 IPC per stage distilled by `PDD` + different base methods. In 'Training' column, U, S, P correspond to training on $\cup_{j=1}^{P} \mathcal{S}_j$, or $\mathcal{S}_i$, or $\cup_{j=1}^{i} \mathcal{S}_j$, at stage $i$, respectively.

| $P$ | Training | Test Accuracy | |
| --- | --- | --- | --- |
| | | `MTT + PDD` | `IDC + PDD` |
| 1 | - | $65.3 \pm 0.7$ | $67.5 \pm 0.5$ |
| 2 | U | $60.4 \pm 0.6$ | $71.1 \pm 0.2$ |
| | S | $64.1 \pm 1.0$ | $68.5 \pm 0.1$ |
| | P | $\mathbf{68.7 \pm 0.8}$ | $\mathbf{71.4 \pm 0.2}$ |
| 3 | U | $65.4 \pm 0.7$ | $\mathbf{74.2 \pm 0.4}$ |
| | S | $67.4 \pm 1.1$ | $68.2 \pm 0.7$ |
| | P | $\mathbf{71.5 \pm 0.4}$ | $74.0 \pm 0.3$ |
| 4 | U | $63.2 \pm 0.7$ | $75.4 \pm 0.3$ |
| | S | $66.0 \pm 0.9$ | $69.9 \pm 0.5$ |
| | P | $\mathbf{73.1 \pm 0.6}$ | $75.4 \pm 0.1$ |
| 5 | U | $65.9 \pm 0.4$ | $76.2 \pm 0.6$ |
| | S | $67.4 \pm 0.8$ | $69.9 \pm 0.5$ |
| | P | $\mathbf{74.2 \pm 0.5}$ | $\mathbf{76.5 \pm 0.2}$ |

In Table 4, we show both components are crucial by comparing the test accuracy of ConvNet after being trained on the `PDD` distilled datasets with both or without one of the two designs. For `PDD + MTT` and both variants, we fix the number of images per class to distill in each stage to be 10. We observe a decreased performance of `PDD` when it distills images for each training stage independent of the previous stages, and the difference is more significant when we distill longer training intervals with more stages.

Table 4: ConvNet's performance on CIFAR-10 with different synthesis modes using `MTT` with `PDD`.

| $P$ | w/o transition | w/o conditioning | `PDD` |
| --- | --- | --- | --- |
| 1 | $65.3 \pm 0.7$ | $65.3 \pm 0.7$ | $65.3 \pm 0.7$ |
| 2 | $66.0 \pm 0.6$ | $67.9 \pm 0.5$ | $68.7 \pm 0.8$ |
| 3 | $66.3 \pm 0.4$ | $69.8 \pm 0.9$ | $71.5 \pm 0.4$ |
| 4 | $65.6 \pm 0.5$ | $71.4 \pm 0.5$ | $73.1 \pm 0.6$ |
| 5 | $63.6 \pm 0.7$ | $71.9 \pm 0.7$ | $74.2 \pm 0.7$ |

Table 5: Models' testing accuracy on CIFAR-10. `PDD` with different numbers of stages ($P$) and per-stage IPC.

| $P$ | per-stage IPC | Accuracy |
| --- | --- | --- |
| 1 | 10 | $65.3 \pm 0.7$ |
| 2 | 5 | $65.5 \pm 0.9$ |
| 5 | 2 | $66.9 \pm 0.4$ |
| 10 | 1 | $64.4 \pm 0.6$ |

**Distilling more training stages vs more images per stage.** Given a fixed total number of images per class, we can distill longer training dynamics by having more stages, or choose to distill more images in each stage to capture the dynamics better. To understand which of the above two strategies leads to better performance, we study four different combinations of the number of stages and per-stage IPC, and record the models' test accuracy in Table 5. We observe that establishing more stages can generally improve the results, as long as per-stage IPC is not too small (IPC = 1 per stage leads to degraded performance). In particular, with 10 as a fixed number of images in total, best result corresponds to $P = 5$ and per-stage IPC = 2.

**Discarding easy-to-learn examples at later stages.** Next, we confirm that samples that are easier to learn can be dropped from the distillation pipeline in later intervals. To do so, we use the forgetting score (Toneva et al., 2019) defined as the number of times the prediction of every example changes from being correctly classified to incorrectly classified during the training. Examples with higher

forgetting scores are more difficult to learn for the network and are learned later during the training (Toneva et al., 2019).

We separate training examples into multiple partitions based on their forgetting scores, with an increment of 3. More specifically, at the $i$-th stage only the examples with a number of forgetting events between $3 \times (i-1)$ and $3 \times i$. Subsequently, we apply PDD to distill the corresponding partition of data examples at each stage, starting from the partition that contains examples with the lowest forgetting scores and progressing to those with the highest scores. Table 6 shows that when PDD explicitly distills examples with increasing learning difficulty at different stages, models trained on the distilled images have the same test performance as when the distillation is based on the full training set at all stages.

Table 6: ConvNet's performance on CIFAR-10 trained on synthetic set with 10 images per class using MTT with PDD by distilling from easy to difficult samples. In $i$-th stage we select samples with forgetting score within $[3(i-1), 3i)$. We report the portion of training samples used in each setting.

| $P$ | Data Used | Testing Accuracy |
|---|---|---|
| 1 | 36.5% | 65.9% |
|  | 100% | 65.3% |
| 3 | 55.8% | 71.6% |
|  | 100% | 71.5% |
| 5 | 66.4% | 73.7% |
|  | 100% | 74.2% |

This observation not only confirms that PDD naturally creates a curriculum with its synthetic sets but also confirms the possibility of reducing the distillation cost of PDD as the training examples used in each stage can be significantly reduced.

## 5.5 CONTINUAL LEARNING

In this section, we adopt a class incremental setting (Zhao et al., 2021; Zhao & Bilen, 2021b) to show that PDD can improve the performance in the application of continual learning. We apply PDD on MTT to distill CIFAR-100 across 5 phases, in each of which we can only access 20

Table 7: Continual learning performance using distilled samples generated by different methods on CIFAR-100.

| Methods | Stage1 | Stage 2 | Stage 3 | Stage 4 | Stage 5 |
|---|---|---|---|---|---|
| DSA | 52.5 | 45.7 | 40.4 | 35.0 | 31.1 |
| Herding | 48.4 | 43.3 | 39.6 | 36.4 | 33.1 |
| MTT | 55.7 | 52.1 | 48.3 | 43.0 | 41.2 |
| **PDD**+MTT | **61.2** | **56.6** | **51.5** | **48.3** | **45.1** |

classes with 20 images distilled in total per class. During the evaluation, a model will be trained sequentially on samples available at each stage. Table 7 shows the performance using different methods, which demonstrates that PDD + MTT consistently outperforms MTT at each stage and showcases PDD's ability to improve baselines' performance in the application of continual learning.

## 5.6 SYNTHESIZED SAMPLES VISUALIZATION

In Figure 4, we provide examples of synthetic samples on CIFAR-10 using PDD + MTT at different stages. We distill CIFAR-10 in 5 stages with a per-stage IPC of 10. From the images we can observe that the synthetic samples at later stages show diversified patterns, demonstrating lower saturation in color and more abstract textures. This evolution of visual patterns indicates a shift in the focus of the distillation process and thus provides an empirical support to our multi-stage design. Figure A5 shows all the samples from Stage 1 to 5 where the transition of distilled patterns on all classes are clearly presented.

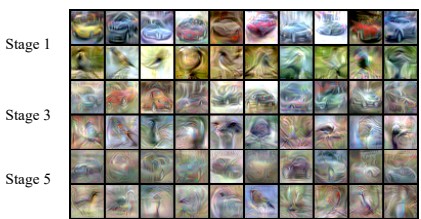

Figure 4: Synthesized images of CIFAR-10 using PDD + MTT from Stage 1, 3 and 5. The images from classes "automobile" and "birds" at each stage are selected for demonstration.

## 6 CONCLUSION

In this work, we proposed a progressive dataset distillation framework, PDD, that generates multiple sets of synthetic samples sequentially, conditioned on the previous ones, to capture dynamics of different training intervals. The multi-stage nature of PDD enables it to capture the training dynamics of neural networks more effectively, and simultaneously reduce the complexity associated with training the entire synthetic dataset. Extensive experiments confirm the effectiveness of PDD in improving the performance of existing dataset distillation methods on various benchmark datasets. For future works, we are interested in exploring PDD's potential to serve as an effective base method for further condensation (Liu et al., 2023a) where the budgets may vary, aligning well with the inherent design of our approach.

## ACKNOWLEDGEMENTS

The work of Y. Yang was supported in part by the Amazon PhD fellowship. The work of B. Mirzasoleiman was supported in part by the U.S. National Science Foundation under the grant IIS-2146492 (CAREER Award). X. Chen and Z. Wang are in part supported by the NSF AI Institute for Foundations of Machine Learning (IFML).

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

## A1    EXPERIMENT DETAILS

### A1.1    EXPERIMENT SETTINGS

On CIFAR-10, the networks are trained for $\frac{2000}{P+1}$ epochs at each stage. Consequently, the total iterations taken is $\frac{P(P+1)}{2} \frac{2000}{P+1} \times \frac{n}{B} = \frac{1000Pn}{B}$, where $B$ is the batch size and $n$ is the number of images newly distilled at each stage. This quantity proves to be adequate in achieving favorable outcomes without inflating the computational burden of network training. Notably, it aligns with utilizing all available images for a training duration of 1000 epochs. Additionally, it is important to note that augmenting the number of epochs could lead to further enhancements in the test accuracy of the trained networks. For CIFAR-100, the networks undergo training for 500 epochs during each stage to facilitate improved convergence.

## A2    MORE VISUALIZATION

In Figure A5 we visualize the synthetic samples of CIFAR-10 distilled at stages 1 to 5 using `PDD + MTT`. We observe a significant shift of visual features in these distill images. The images distilled at the first stage are the most colorful among all the distilled samples, while the images distilled at later stages contain more abstract features and less focus on colours. These figures show that `PDD` helps distill diverse features according to different stages.

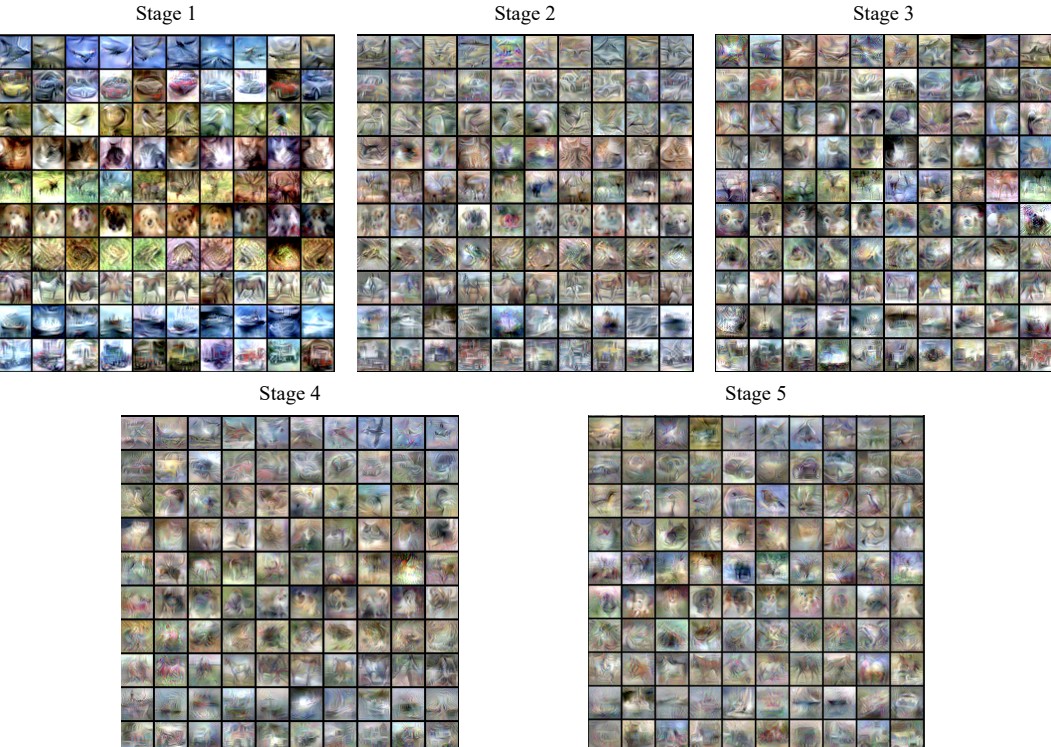

Figure A5: Visualization of synthesized samples from Stage 1 to Stage 5.

## A3  MORE EXPERIMENT RESULTS

### A3.1  RESULTS ON MORE METHODS

**Experiments on DC, DSA and DM.** We further apply PDD to DC (Zhao et al., 2021) and DSA (Zhao & Bilen, 2021b) to distill images from CIFAR-10. Table A8 shows the ConvNet's accuracy after trained on the distilled images. On DC and DSA, compared to using the single stage synthesis, PDD + DC and PDD + DSA generates samples that lead to higher performance, improving the baselines' performance by 2.4% and 0.7%, respectively.

Additionally, we have conducted experiments with DM on CIFAR-10 with IPC=50. Note that DM does not distill the training dynamics but rather enforces the similarity between features extracted from synthetic and real samples by enormous models with random weights. Our experiments show that for IPC=50, DM hits an accuracy of 63.0% while PDD + DM can reach 63.4%, showing a slight 0.4% improvement in performance. This mild improvement is expected, as DM does not capture any training dynamics of training on full data, and only captures similarity between features based on random weights.

Table A8: ConvNets' test accuracy on CIFAR-10 after trained on synthetic samples generated by DC, DSA and DM with different numbers of images per class.

| Dataset IPC | CIFAR-10 50 |
|---|---|
| DC | $53.9 \pm 0.5$ |
| DSA | $60.6 \pm 0.5$ |
| DM | $63.0 \pm 0.4$ |
| PDD + DC | $\textbf{56.3} \pm 0.5$ |
| PDD + DSA | $\textbf{61.3} \pm 0.4$ |
| PDD + DM | $\textbf{63.4} \pm 0.5$ |
| Full | 88.1 |

Table A9: ConvNets' test accuracy on CIFAR-10 after trained with samples distilled by DREAM and DREAM+PDD.

| Dataset IPC | CIFAR-10 10 | 50 |
|---|---|---|
| Accuracy of DREAM | 68.7 | 74.8 |
| Accuracy of PDD + DREAM | 69.3 | 76.7 |
| Improvement of PDD | **0.6** | **1.9** |
| Standard Deviation | 0.3 | 0.2 |
| Full | 88.1 | |

**Experiments on DREAM.** We have conducted an additional set of experiments applying PDD to DREAM on CIFAR-10. The results using the IPC values of 10 and 50 are presented in Table A9, where we can observe that PDD can further improve the performance of DREAM beyond the current state-of-the-art method.

### A3.2  EXPERIMENTS ON UPDATING PREVIOUS FROZEN SUBSETS

We have conducted an additional set of experiments that allows images synthesized at early stages to be updated during late stages with a smaller learning rate. We choose MTT as the base method and set up 5 stages with a per-stage IPC of 10, resulting in a total IPC of 50. During the evaluation phase, we ensure the fairness of comparison by only using the last set of synthesized images to train models. This is to keep the number of synthesized samples to be the same when compared with other results. The trained model hits a test accuracy of 69.1%, which is inferior to our proposed method.

The decline in performance can likely be attributed to the fact that expert trajectories, i.e, the subsequent model checkpoints obtained by training on the full real dataset and synthetic samples (if applicable, from previous stages), are not derived from a consistent set of synthetic samples. Note that we first train on synthetic examples produced in previous stages to generate expert trajectories and synthesize data for the next stage. If the synthesized images of the previous stages are updated, the distilled samples from stage 1, which serve as the foundation for generating trajectories in stages 2 to 5, vary with each subsequent stage. When the samples synthesized in stage 1 are updated in stage 2, they gradually deviate from the initial synthetic samples (i.e., those employed to generate the expert trajectories for stage 2). Hence, training on the synthesized images that are being updated results in a different training dynamics each time. This eventually leads to discrepancy and ultimately results in a decrease in performance.

