# OpenReview forum: "Data Distillation Can Be Like Vodka: Distilling More Times For Better Quality"
_ICLR.cc/2024/Conference — ICLR 2024 poster_

### Official Review · Reviewer_xT2K · 2023-10-29

**Soundness:** 3 good
**Presentation:** 3 good
**Contribution:** 3 good
**Rating:** 8
**Confidence:** 3

**Summary:**

The significant contribution of this work lies in the introduction of Progressive Dataset Distillation (PDD). It is a novel methodology in the field of dataset distillation. PDD effectively addresses the limitations inherent in existing DD works by recognizing that relying solely on a single synthetic subset for distillation does not lead to optimal generalization.

Specifically, PDD innovatively synthesizes multiple small sets of synthetic images, with each set being conditioned on the knowledge acquired from the preceding sets. The alteranated updated model is then trained on the cumulative union of these subsets. This sophisticated approach results in a noteworthy performance improvement.

**Strengths:**

1. The idea presented in this work is both intriguing and groundbreaking. As far as my knowledge extends, this is the first instance where the synthesis of multiple small sets of distilled images has been proposed. The authors have adeptly addressed the challenge of minimizing bias and ensuring training stability by introducing a conditioning mechanism for each distilled set, based on the knowledge accumulated from the preceding sets.

2. The clarity and coherence of the writing are commendable. The motivation behind the research is robust, and the overall structure of the paper is meticulously organized.

3. The experimental results provided in the paper are highly promising and effectively illustrate the efficacy of the proposed solution.

**Weaknesses:**

N.A.

**Questions:**

Is it possible to provide additional experimental results about "DM+PDD"?

---

> ### Author Response · Authors · 2023-11-21
> **Response to Reviewer xT2K**
>
> Thank you for your positive feedback on our work and for recognizing its novelty and significance in the field of dataset distillation. We appreciate your comments on the clarity of our writing and the robustness of our research motivation.
>
> **1. Additional Experimental Results on "DM+PDD":**
> Certainly! We have conducted experiments with DM on CIFAR-10 with IPC=50. Note that DM does not distill the *training dynamics* but rather enforces the similarity between features extracted from synthetic and real samples by enormous models with *random weights*. This is why we did not consider DM initially in our experiments, as our goal is to capture *longer-term training dynamics on full data*. Our experiments show that for IPC=50, DM hits an accuracy of 63.0% while DM+PDD can reach 63.4%, showing a slight 0.4% improvement in performance. This mild improvement is expected, as DM does not capture any training dynamics of training on full data, and only captures similarity between features based on random weights.

---

### Official Review · Reviewer_rp5C · 2023-10-31

**Soundness:** 3 good
**Presentation:** 3 good
**Contribution:** 3 good
**Rating:** 5
**Confidence:** 4

**Summary:**

The paper proposes a dataset distillation method. The core idea is to freeze the previously distilled subset and then only optimize the new subset. This method can be applied to several dataset distillation methods and show marginal improvement over the original method.

**Strengths:**

1. The overall writing is clear.
2. The comparison with a few previous is clear to validate the effectiveness of the method.
3. The explanation looks sufficient.

**Weaknesses:**

1. Some important reference is missing such as DEARM [a]. Since this is the SOTA method in dataset distillation, the authors should cite and compare the paper.
2. After Eq.4, the authors claim PDD can be used to **any** dataset distillation method. Therefore, it is necessary to do experiment if the PDD can augment DREAM.
3. The accuracy is not as good as DREAM. For example, on CIFAR-10 IPC-10, DREAM can achieve 69.4% accuracy. But the proposed PDD can just achieve 67.9%. On CIFAR-100 IPC-10, DREAM can achieve 46.8% accuracy. But the proposed PDD can just achieve 45.8%.
4. Based on question 3, why does the proposed method have bad performance on IPC-10?
5. The idea is too straightforward. Is it possible to update the previous frozen subset with small learning rate?


 I will increase the score if the DREAM-related experiments are added.

[a] DREAM: Efficient Dataset Distillation by Representative Matching, ICCV 2023

**Questions:**

see weakness

---

> ### Author Response · Authors · 2023-11-21
> **Response to Reviewer rp5C**
>
> We appreciate the reviewer's feedback and recognition of the clarity and effectiveness of our method. We address the specific concerns and suggestions as follows:
>
> **1. Missing Reference to DREAM [a]:**
> Thank you for pointing out the missing reference. However, we would like to highlight that our submission to ICLR is concurrent with the public release (Aug 2023) of DREAM’s ICCV version. This timing meant we were not aware of the advancements presented in DREAM by the time we submitted our work.
>
> Per your request, we have:
> 1. included and briefly discussed DREAM [a] in the revised manuscript (in Section 2; updates are highlighted in pink).
> 2. conducted experiments using DREAM [a] as the base method on CIFAR-10 (please refer to Question 2, and also the appendix).
>
> However, this timing and the short period of rebuttal still result in a constraint on our ability to conduct additional experiments on CIFAR-100 and Tiny-ImageNet. We are committed to extending our research and will diligently work to incorporate these datasets into our analysis. We will share these expanded results as soon as they are available.
>
> [a] DREAM: Efficient Dataset Distillation by Representative Matching, ICCV 2023
>
> **2. Experiment with DREAM as the Base Method:**
> Certainly! We have conducted a set of experiments with DREAM as the base method on CIFAR-10 with IPC values of 10 and 50, respectively. The results are summarized in the table below. Using their provided implementation, we observed a slightly lower performance of DREAM, compared to what is reported in their paper (which might be due to the difference between computational devices and environments). However, using the same implementation and computational framework, we see that PDD can further improve the performance of DREAM beyond the current SOTA. These results have also been included in the Appendix and will be further incorporated into the main text.
>
> | Method | IPC=10 | IPC=50 |
> | :-: | :-: |  :-: |
> | Accuracy of DREAM (reproduced) | 68.7 | 74.8 |
> | Accuracy of DREAM + PDD | 69.3 | 76.7 |
> | Improvement of PDD | 0.6 | 1.9 |
> | Standard Deviation | 0.3 | 0.2 |
>
> **3 & 4. Performance Comparison with DREAM**
> We respectfully disagree that our method has a poor performance when IPC equals 10 due to the following reasons:
>
> 1. Firstly, we argue that it is not a fair comparison to contrast the accuracy of DREAM with the numbers in Table 1, as they deploy MTT or IDC as base methods.
> 2. Secondly, our method consistently improves the performance of base methods with substantial performance gaps. Specifically, PDD substantially improves the performance of IDC and MTT on CIFAR-10 by 0.4% and 1.6%, respectively. These improvements in performance are also acknowledged by Reviewer xT2K and S7ej.
> 3. Moreover, it is important to note that we only observe a smaller improvement for the combination of PDD and IDC on CIFAR-10 when IPC equals 10. This is likely due to IDC’s inability to distill longer training trajectories on real images (merely 4 epochs). Consequently, even extending to multiple stages is not enough for distilling more difficult images. For example, setting 5 stages approximately distills merely 20 epochs on real images, which is deemed insufficient. This is likely a limitation embedded in IDC, yet not in our PDD method.
> 4. Finally, the additional experiment results we provided in Question 2 show that PDD can also improve the performance of DREAM on CIFAR-10 with IPC=10 and 50.
>
> Finally, we reiterate that our contribution lies in the revelation that a multi-stage synthesis approach can enhance the ability of base methods to capture training dynamics more effectively when compared to their single-stage counterparts. While distilling different subsets for different stages of training may seem intuitive, **it has not been previously explored**. **We are the first to introduce this multi-stage synthesis framework, which is acknowledged as “intriguing and groundbreaking” by Reviewer xT2K and “interesting” by Reviewer S7ej.**
> More importantly, our framework not only improves performance on standard IPCs, such as 10 and 50, but also represents **a unique advance by enabling the generation of a large number of samples for the first time**. This can equip any base method with the capability to significantly reduce the performance gap compared to training with complete datasets.

---

> > ### Comment · Reviewer_rp5C · 2023-11-21
> >
> > Thanks for your effort in responding.
> > 1. DREAM [a] was publicly released on Feb. 2023 on Arxiv and was also open-sourced then. This is half a year before their ICCV version.
> > 2. The authors fail to show the experiments on CIFAR-100 and ImageNet. So, I am slightly doubtful about the proposed method's scalability.
> > 3. The improvement over DREAM is marginal—just 0.6% on IPC-10. It shows the power of the proposed method to some extent.
> > 4. If the authors provide accuracy improvement on **CIFAR-100 on IPC-10**, which does not require much computation resources. I will raise my score to 5.
> >
> >
> >
> > [a] DREAM: Efficient Dataset Distillation by Representative Matching, ICCV 2023

---

> > > ### Author Response · Authors · 2023-11-23
> > > **Additional Response to Reviewer rp5C**
> > >
> > > We thank the reviewer for the feedback. We provide a point-to-point response to your concerns.
> > >
> > > **AQ1: The scalability of our method?**
> > > Indeed, one of the most striking advantages of our method is its scalability:
> > > We have provided results on CIFAR-100 and TinyImageNet in Table 1 with IPC=10 and 50. We have shown that the performance can be improved on these datasets under both settings.
> > > We have provided results on CIFAR-10 and CIFAR-100 with IPC that are greater than 50, presented in Figure 3. More concretely, we achieve an IPC of **250** on CIFAR-10, an IPC of **100** on CIFAR-100, using PDD with MTT as base methods. To our knowledge, this has been made possible for the **first time** by our method.
> > > These results on CIFAR-100 and TinyImageNet, and also larger numbers of IPC beyond current common practice, collectively demonstrate the scalability of our framework.
> > >
> > > **AQ2: Marginal improvement on IPC=10?**
> > > We believe that the reason is that the trajectories are relatively short, merely $4$ epochs on real data. Therefore, having even $5$ stages results in $20$ epochs in total, which might not be sufficient for large improvement when IPC=10. However, when IPC=50, PDD further improves the performance by $1.9%$ if applied to DREAM. This is because training the network on a larger number of synthetic images in each stage can improve the performance of the model much more effectively. This has a similar effect to distilling and training on synthetic data generated based on longer trajectories of training on real data. When trajectories are longer or IPC is larger, synthetic data generated in the next stage captures training dynamics of later phases in training more effectively (this is not the case when trajectories and IPC are both small). On another base method, MTT, PDD exhibits more significant improvement even when IPC=10, owing to the longer training trajectories (15 epochs) adopted by MTT. This confirms our argument and is also explained in our responses to **Q3 & 4**.
> > >
> > > **AQ3: Improvement over DREAM on CIFAR-100 with IPC=10?**
> > > We have followed your suggestions to conduct experiments on CIFAR-100 with IPC=10.
> > > We have observed an improvement of 0.2% of test accuracy over DREAM, again demonstrating the effectiveness of our algorithm.

---

> > > ### Author Response · Authors · 2023-11-23
> > > **Additional Response to Reviewer rp5C (cont'd)**
> > >
> > > **AQ4: Publish date of DREAM and comparison with contemporaneous works.**
> > >
> > > Dear Reviewer rp5C,
> > >
> > > We checked the ICLR Reviewer Guideline (https://iclr.cc/Conferences/2024/ReviewerGuide) once again, and we noticed at the bottom of the guideline that **only papers that are published (published means they are available in online proceedings) before May 28, 2023 are required to be compared with the proposed method**. While DREAM was indeed online back in February, it was not peer-reviewed at that time, making it fall into the category of contemporaneous works which we did not have to compare with.
> > >
> > > We acknowledge the contribution of DREAM, and we have made additional experiments to compare our method with it. We will add the full experiments with larger IPCs to our revised version, which we expect to yield larger improvements as we explained in our answer (we did not have enough time to finish experiments with larger IPC during the discussion phase). We hope this can address your remaining concern.
> > >
> > > Best,
> > >
> > > Authors

---

> > > > ### Comment · Reviewer_rp5C · 2023-11-23
> > > >
> > > > Thanks for your response. I will raise my score.

---

> > > > > ### Author Response · Authors · 2023-11-23
> > > > >
> > > > > Dear Reviewer rp5C,
> > > > >
> > > > > We are truly grateful for your time and support and we are glad to see our responses have addressed your concerns. Thank you for raising the score.
> > > > >
> > > > > Best wishes,
> > > > >
> > > > > Authors

---

> ### Author Response · Authors · 2023-11-21
> **Response to Reviewer rp5C (cont'd)**
>
> **5. Idea of Updating Previous Frozen Subset:**
> We emphasize that our approach, while simple, is demonstrably effective and can be seamlessly integrated with any base method to enhance performance and scalability. This strength has been acknowledged by Reviewer S7ej.
>
> Regarding the suggestion to update the previously frozen subset with a small learning rate, it is important to note that such an approach would void the scalability (i.e. generating larger synthetic data) enabled by our method since the samples distilled at previous stages need to be further optimized at later stages, requiring additional computational cost and memory consumption.
>
> Nevertheless, we have conducted an experiment that allows images synthesized at early stages to be updated during late stages with a smaller learning rate (i.e., 10% of the original learning rate). We choose MTT as the base method and set up $5$ stages with a per-stage IPC of $10$, resulting in a total IPC of $50$. The trained model hits a test accuracy of 69.1%, which is inferior to our proposed method.
>
> The decline in performance can likely be attributed to the fact that expert trajectories, i.e., the subsequent model checkpoints obtained by training on the full real dataset and synthetic samples (if applicable, from previous stages), are not derived from a consistent set of synthetic samples. Note that we first train on synthetic examples produced in previous stages to generate expert trajectories and synthesize data for the next stage. If the synthesized images of the previous stages are updated, the distilled samples from stage 1, which serve as the foundation for generating trajectories in stages 2 to 5, vary with each subsequent stage. When the samples synthesized in stage 1 are updated in stage 2, they gradually deviate from the initial synthetic samples (i.e., those employed to generate the expert trajectories for stage 2). Hence, training on the synthesized images that are being updated results in different training dynamics each time. This eventually leads to discrepancy and ultimately results in a decrease in performance. The above discussion has been incorporated into our manuscript.

---

### Official Review · Reviewer_S7ej · 2023-11-01

**Soundness:** 2 fair
**Presentation:** 2 fair
**Contribution:** 2 fair
**Rating:** 6
**Confidence:** 5

**Summary:**

This paper proposes a dataset distillation algorithm that generates synthetic data in a progressive manner: the next batch of synthetic data would be dependent on previous batches. The training using the distilled datasets also contains several stages. The training data for each stage come from the corresponding batch of the distilled dataset and its previous batches in a cumulative way. Experiments demonstrate that the proposed strategy improves the baseline method.

**Strengths:**

The idea of progressive dataset distillation is interesting. This can 1) capture the training dynamic of neural networks better as demonstrated by authors, 2) reduce the complexity of training the whole synthetic dataset together, and 3) serve as a strong method for slimmable dataset condensation [a].

[a] Slimmable Dataset Condensation, CVPR 2023.

**Weaknesses:**

1. Many places are unclear.
    * Fig. 1 needs some explanations. How do the results come in detail, like what's the IPC of each stage, and how to conduct multi-stage training? Although some of these questions are answered in the following parts, the writing is not coherent.
    * The networks for the next stage come from the training results with previous batches of synthetic data. In IDC, the networks are periodically re-initialized randomly. In MTT, the networks come from checkpoints of training with original datasets. We have to conduct some modifications to these baselines before using PDD. These operations are unclear.
2. A comment: this method makes an assumption on downstream training using synthetic datasets: models must be trained in a multi-stage way using the provided multi-stage synthetic data, which would introduce a lot of hyperparameters, especially in the cross-architecture setting and make the dataset distillation less elegant. Given that the performance gain is not significant in most cases, the practical value of the proposed method is somewhat limited.
3. Through the results in Tab. 6, the effect of discarding easy-to-learn examples at later stages is not significant. More evidence is necessary to demonstrate the effectiveness.

**Questions:**

Please refer to the Weaknesses part. I would also like the authors to discuss more benefits of PDD as mentioned in the 1st point of Strengths in the camera-ready version or in the next revision cycle.

---

> ### Author Response · Authors · 2023-11-21
> **Response to Reviewer S7ej**
>
> We sincerely thank the reviewer for the constructive feedback. We address your concerns point-to-point below.
>
> **1. Further explanation:**
> We’d like to note that the details of our method are accurately specified in the paper. We will further elaborate below:
>
> *1.1. Explanation of PDD for Figure 1:*
> Thank you for your feedback regarding Figure 1. This Figure is mainly the motivation of our method, and hence the details are specified later. To enhance the clarity, we have updated the caption of Figure 1 to: “Our proposed multi-stage dataset distillation framework, PDD, improves the state-of-the-art algorithms by iteratively distilling smaller synthetic subsets that capture longer training dynamics on full data. In the setting shown in the figure, PDD uses MTT as its base distillation method to incrementally generate $5$ synthetic subsets of size $10$, where each subset captures $15$ epochs of training on full data.“
>
> *1.2. Clarification about Base Method Modification*
> We clarify that PDD seamlessly integrates with the base methods with hardly any modification and without introducing any additional hyperparameters (which is why our paper does not include discussion on extra hyperparameters):
>
> For the first stage, we distill datasets using the base methods in their original form.
> Starting from the second stage, we distill with the same base algorithm. However, the model weights employed in these stages are those that have been trained on examples distilled in the earlier stages. For instance, we train the newly sampled networks with previous synthesized samples for IDC, and we start new collections of training trajectories from weights trained with previous samples for MTT.
> **These operations are detailed in our pseudocode and also formulated in Equation (4)**. This ensures a smooth integration of PDD with existing frameworks, enhancing their effectiveness without the need for fundamental modifications to the base methods.
>
> **2. Practicality of Multi-Stage Training:**
> Our method does not introduce additional hyperparameters: IPC per stage and the number of stages are determined based on the available budget, and PDD inherits all the rest of hyperparameters, such as learning rate, weight decay, number of training steps, etc., from the base method. This ensures that the practicality of our method is maintained by leveraging existing parameters, thereby simplifying the adoption and integration of PDD in various settings.
>
> Furthermore, we wanted to emphasize that our method brings significant performance gain for the following reasons:
> 1. Our results have already outperformed current state-of-the-art methods in multiple scenarios, even when using less powerful base methods. For instance, on CIFAR-10 with an IPC of 50, our approach combining PDD and IDC surpasses DREAM, the current SOTA by 1.7%. Similarly, on CIFAR-100 with an IPC of 50, PDD + IDC outperforms DREAM by 0.5%. Notably, PDD can be also easily combined with DREAM to improve its performance and achieve SOTA, as we will report in our response to reviewer rp5C.
> 2. Across other settings, our methods consistently enhance the performance of the base methods, approaching the performance level of the current state-of-the-art benchmarks.
>
> **3. Discarding Easy-to-Learn Examples:**
> Discarding difficult-to-learn examples is very effective in improving the efficiency of our method, and also improves its performance for smaller values of P. Table 6 shows that around 2/3 of examples can be discarded in the first stage and around 2/3 of examples can be discarded in the second and third stages. That is, only 1/3 of the (easiest-to-learn) data is required for the first stage and another 1/3 (easy but not easiest examples) is required for the second and third stages. Discarding difficult-to-learn examples not only considerably improves the distillation efficiency, but also improves the performance of dataset distillation, especially in earlier stages. This is because easy-to-learn examples contribute the most to learning in the early stages. Hence, focusing the distillation on such examples in earlier stages improves both the efficiency and the quality of the generated synthetic examples, as is evident in Table 6, P=1 and P=3.

---

> > ### Comment · Reviewer_S7ej · 2023-11-22
> >
> > Thanks to the authors for the response to my concerns. I think this time the contribution and technical details are better clarified. Most of my concerns are alleviated. Currently, I still think Fig. 1 is misleading because the total IPC of PDD is 50 but the baseline is 10, which is an unfair comparison. It would be better if the authors could also mark the accuracy of 20, 30, 40, and 50 IPC on the plot. At least 50 IPC is necessary for a fair comparison.
> >
> > Overall, I choose to raise my score to 6.

---

> > > ### Author Response · Authors · 2023-11-23
> > > **Additional Response to Reviewer S7ej**
> > >
> > > Thank you for your comments. We are glad to see that our responses have addressed your comments. We have followed your suggestion to add the accuracy of IPC=20,30,40,50 on Figure 1 and update the x-axis to avoid misunderstanding.

---

> ### Author Response · Authors · 2023-11-21
> **Response to Reviewer S7ej (cont'd)**
>
> **4. More Discussion on Benefits of PDD:**
> Thank you for the suggestion to elaborate on the benefits of PDD in our manuscript. In the revised version, we discussed the advantages of our method again in both the introduction and conclusion sections (highlighted in pink). Specifically, our original introduction highlights that PDD captures the training dynamics of neural networks over longer intervals more effectively. We have added the following suggestion by the reviewer to our introduction: Importantly, PDD reduces the complexity associated with training the entire synthetic dataset simultaneously, and can serve as an effective base method for slimmable dataset condensation [a] which handles changes in the budget for storage or transmission.
>
> We hope that these revisions and clarifications will address the concerns raised and demonstrate the value of our proposed PDD method. We appreciate the opportunity to improve our manuscript and thank the reviewer for their valuable input.
>
> [a] Slimmable Dataset Condensation, CVPR 2023.

---

### Author Response · Authors · 2023-11-21
**Summary of Update and Rebuttal**

We extend our sincere gratitude to all reviewers for dedicating their time and expertise to reviewing our manuscript.

In response to the valuable feedback from Reviewer S7ej, we have enriched the introduction and the conclusion of our paper with a detailed discussion on the advantages of the PDD approach.
Following the insightful suggestions by Reviewer rp5C, we have undertaken additional experiments focusing on DREAM. The results of these experiments have been included in the appendix for a comprehensive review, and the full version of these results will be eventually incorporated into the main paper. Moreover, we have added a paragraph in our paper to elaborate on this SOTA method. Additionally, the experimental findings concerning DM, as suggested by Reviewer xT2K, have been incorporated into the appendix as well. The updates have been marked in pink in the revision.

Due to the limited time of rebuttal, we provide results on CIFAR-10 for all the additional experiments only. We will work diligently and continuously to obtain results on other datasets (CIFAR-100, TinyImageNet), and we are committed to incorporating them in the final manuscript.

We have taken careful consideration to address each query raised by the reviewers in our responses. Should there be any further questions or clarifications needed, we are more than willing to provide additional information and engage in further discussion.

---

### Meta-Review · Area_Chair_Q7Ef · 2023-12-05

**Metareview:**

The paper presents a novel dataset distillation algorithm, Progressive Dataset Distillation (PDD), which introduces a multi-stage synthetic data generation approach where each batch depends on the previously distilled data, cumulatively used for training. Reviewers generally agree that the idea is innovative and has the potential to capture neural network training dynamics more effectively, reducing complexity and potentially serving as a robust method for dataset condensation. However, they raised concerns mainly about the clarity of the methodology, the sufficiency of experimental comparison, and the practical value due to limited performance gains and the introduction of additional hyperparameters in a cross-architecture setting.

During the rebuttal, the authors presented several important clarifications, including (1) their method does not introduce additional hyperparameter and will inherit all the rest of hyperparameters of the base method; (2) adding discussions and experimental comparison to DREAM (ICCV'23), showing their method can still consistently improve the performance. Those have been acknowledged by reviewers in follow-up discussions and have cleaned up most concerns. Therefore, the paper's contribution is seen as solid and, with revision and integration of rebuttal contents, could warrant acceptance.

**Justification For Why Not Higher Score:**

Reviewers raised concerns mainly about the clarity of the methodology, the sufficiency of experimental comparison, and the practical value due to limited performance gains and the introduction of additional hyperparameters in a cross-architecture setting.

**Justification For Why Not Lower Score:**

Reviewers generally agree that the idea is innovative and has the potential to capture neural network training dynamics more effectively, reducing complexity and potentially serving as a robust method for dataset condensation

---

### Decision · Program_Chairs · 2024-01-16

Accept (poster)